# Vitamin–Microbiota Crosstalk in Intestinal Inflammation and Carcinogenesis

**DOI:** 10.3390/nu14163383

**Published:** 2022-08-17

**Authors:** Zihan Zhai, Wenxiao Dong, Yue Sun, Yu Gu, Jiahui Ma, Bangmao Wang, Hailong Cao

**Affiliations:** Department of Gastroenterology and Hepatology, Tianjin Medical University General Hospital, Tianjin 300052, China

**Keywords:** vitamins, microbiome, IBD, CAC, therapy

## Abstract

Inflammatory bowel disease (IBD) and colitis-associated colorectal cancer (CAC) are common diseases of the digestive system. Vitamin deficiencies and gut microbiota dysbiosis have a close relationship with the risk, development, and progression of IBD and CAC. There is a strong link between vitamins and the gut microbiome. Vitamins are extremely crucial for maintaining a healthy gut microbiota, promoting growth and development, metabolism, and innate immunity. Gut microbiota can not only influence the transport process of vitamins, but also produce vitamins to compensate for insufficient food intake. Emerging evidence suggests that oral vitamin supplementation can reduce inflammation levels and improve disease prognosis. In addition, improving the diet structure and consuming foods rich in vitamins not only help to improve the vitamin deficiency, but also help to reduce the risk of IBD. Fecal microbiota transplantation (FMT) and the application of vitamin-producing probiotics can better assist in the treatment of intestinal diseases. In this review, we discuss the interaction and therapeutic roles of vitamins and gut microbiota in IBD and CAC. We also summarize the methods of treating IBD and CAC by modulating vitamins. This may highlight strategies to target gut-microbiota-dependent alterations in vitamin metabolism in the context of IBD and CAC therapy.

## 1. Introduction

IBD is an idiopathic chronic inflammatory disease, including Crohn’s disease (CD) and ulcerative colitis (UC) [1]. At present, the incidence and prevalence of IBD are on the rise; this is accompanied by numerous extraintestinal manifestations [2]. Chronic inflammation may be conducive to the occurrence or development of tumors. The underlying mechanism is repeated stimulation of epithelial cells and infiltration of various immune cells. According to the International Agency for Research on Cancer (IARC) research, gastrointestinal cancers represented one of the leading causes of mortality worldwide [3]. Colorectal cancer (CRC) is the second most common cause of cancer death worldwide; in addition, morbidity and mortality are still on the rise [4,5]. Many data indicate that patients with IBD are at a higher risk of developing CRC [6]; moreover, 15% of deaths in IBD patients are associated with CAC [7], which supports the notion that inflammation plays a key role in cancer development [8].

Gut microbiota refers to the collection of microorganisms present within the digestive tract [9]. Their function is to protect the integrity of the intestinal epithelial barrier and inhibit the expansion of pathogenic bacteria [10]. Therefore, homeostasis of the gut microbiota is essential. However, many common substances can affect the gut microbiota, such as food additives, nitrosamines, antibiotics, etc. [11]. Meanwhile, the intestinal microbiota also affects the human metabolism by producing enzymes not encoded in the human genome; these include decomposing polysaccharides, polyphenols, and synthesizing vitamins [12]. Through the study of animals and human volunteers, it is found that the intestinal microbiota can synthesize vitamin K, vitamin C, and B vitamins, including biotin, cobalamin, folic acid, niacin, pantothenic acid, pyridoxine, riboflavin, and thiamine [13]. These vitamins have a protective effect on the intestine. Therefore, an imbalance in intestinal microbiota composition may promote the development of inflammation and tumors by affecting vitamin metabolism.

Vitamins are a type of micronutrient that humans and animals must obtain from food in order to maintain normal physiological functions. They play a vital role in the growth, metabolism, and development of the human body. According to the solubility of vitamins, they are divided into water-soluble and fat-soluble; fat-soluble vitamins include vitamin A, D, E, and K. Humans obtain vitamins mainly from the diet, and the intestine is the main absorption site. For example, vitamin A is mainly assimilated in the proximal jejunum, vitamin D is optimal assimilated in the distal jejunum, and vitamins E and K are mainly taken in the ileum [14]. Therefore, impairment of intestinal function may affect the absorption of vitamins. Vitamins can directly or indirectly regulate the intestinal microbiota. For example, vitamins B and D may be involved in the remodeling of the gut microbiota in normal and obese individuals [15]. More and more evidence shows that a vitamin deficiency will lead to an intestinal flora disorder, and then cause intestinal diseases. A deficiency of gut microbes that synthesize B vitamins may lead to the worsening of CD [16]. Even so, dysplastic gut microbiota and abnormalities in B vitamin metabolism have been observed in malnourished children [17]. However, the mechanism of vitamin–microbiota interaction in intestinal inflammation and carcinogenesis is unknown. In recent years, research interest in the role of vitamin–microbiota crosstalk in intestinal inflammation and carcinogenesis has surged. This review describes the mechanism of vitamins in maintaining the homeostasis of the gastrointestinal microbiota and reducing intestinal inflammation to prevent cancer; moreover, it introduces the application of vitamins in the treatment of intestinal diseases. This provides a new idea for the clinical prevention and treatment of IBD and CAC.

The type of this review is a narrative review. We searched mainly in the PubMed database and Web of Science database. The key words are vitamins, intestinal flora, inflammatory bowel disease, colitis-related colon cancer, probiotics, fecal microbial transplantation, etc. We summarize the key literature, and put forward our own understanding and outlook; we thus form this review. We objectively selected the most relevant literatures on this topic and summarized them.

## 2. Interaction of Vitamins and Gut Microbiota

### 2.1. Gut Microbiota Changes in IBD and CAC

The disturbance of gut microbiota plays an important role in the occurrence and development of IBD and CAC. In CAC, the abundance of *Fusobacterium*, *Ruminococcus* genus decreased and the abundance of *Enterobacteriacae* family, *Sphingomonas* genus increased in the gut microbiota [18]; whereas in IBD, the number of *Lactobacillus* spp., *Bifidobacterium* spp., *F. prausnitzii* decreased and the number of *E. coli*, *C. difficile* increased [19]. The decrease in the diversity of *Firmicutes* is a major change in gut microbiota dysbiosis in IBD patients. *Firmicutes* are the main bacteria that synthesize vitamins in the gut microbiome; thus, this may help explain the prevalence of vitamin deficiencies in IBD patients [20,21,22].

### 2.2. The Relationship between Vitamins and the Gut Microbiota

Vitamins are important micronutrients for the human body. Although the human body cannot synthesize vitamins, the gut microbiota can produce vitamins to compensate for insufficient food intake, and acts as a key mediator of vitamin absorption. For instance, vitamin B5 can be obtained from food or synthesized by intestinal flora. Fat-soluble vitamins alter gut microbiota composition by modulating immunity, bacterial growth, and metabolism. Vitamin D supplementation in infant diets has important effects on changes in early microbial composition, whereas vitamin D deficiency in children leads to reduced bacterial diversity. At the same time, gut microbiota can also affect the synthesis, metabolic process, and transport of fat-soluble vitamins and their metabolites [23,24,25,26,27]. Unlike vitamins obtained from food, vitamins produced by microorganisms are mainly absorbed in the colon [28].

### 2.3. The Ability of the Gut Microbiota to Produce Vitamins

Genome annotations, a new method, can forecast vitamin metabolism pathways and evaluate vitamin biosynthesis potential. The author successfully predicted that 40–65% of human gut microbes have the power to synthesize B vitamins. The two most commonly synthesized vitamins are riboflavin and niacin, with 166 and 162 predicted producers, respectively [29]. In a study comparing the taxonomic characteristics of metabolic genes related to vitamin biosynthesis, and transport at the phylum or species level, researchers used these mapped gene identifiers to retrieve the UniProt database’s lineage information. They detected that Firmicutes is the cardinal correlative metabolic pathway of vitamins, accounting for nearly half of the total; the second is Proteobacteria, accounting for nearly a fifth; followed by Bacteroides and Actinomycetes, accounting for less [30]. These four florae are the main components of the human gut microbiota, accounting for more than 90% of the total microbiota.

### 2.4. The Effect of Vitamins on the Gut Microbiota

Correspondingly, vitamins will affect the composition of the gut microbiota. A study investigated 96 healthy volunteers; these volunteers took different vitamins, including vitamin A, vitamin B2, vitamin C, vitamin B2 + C, vitamin D3 and vitamin E, or placebo every day for a period of four weeks. The results showed that: vitamins B2, C, and D affect the metabolism and composition of the gut microbiota; vitamin C prominently raises the alpha diversity of gut microbes; the number of species of gut microbes markedly increased after vitamin B2 supplementation; combined supplementation with vitamins B2 and C resulted in a significant reduction in *Sutterella*, but an increase in *Coprococcus* numbers; and after taking vitamin D, the growth of *Actinobacteria* is promoted and the growth of *Bacteroidetes* is inhibited [31]. Vitamin A deficiency decreased the numbers of the *Lachnospiraceae _NK4A136 _group*, *Anaerotruncus*, *Oscillibacter* in *Firmicutes*, and also decreased the level of *Mucispirillum*; however, *Parasutterella* showed an upward trend. TLR4 may be involved in the process of vitamin A regulating the microbiota [32]. Vitamin C supplementation reduces the number of *Enterobacteriaceae*, increases the abundance of *Lactobacillus* sp., inhibits the growth of harmful bacteria, and promotes the increase of beneficial bacteria [33]. Interaction of vitamin D and the gut microbiota is critical for immune homeostasis. Supplementation with high levels of Vitamin D increased *Prevotella* and decreased *Veillonella* and *Haemophilus*. In addition, *Coprococcus* and *Bifdobacterium* were negatively correlated with 25(OH)D levels [34]. An animal experiment manifested that *Ruminococcus*, *Lachnospiraceae*, and *Muribaculaceae* are more abundant in the gut of vitamin K deficient mice. However, *Bacteroides* is abundant in the vitamin supplement mice. Surprisingly, this phenomenon was more pronounced in female mice than in male mice [35].

Effects of vitamin supplementation on the human gut microbiota are shown in Table 1.

## 3. The Role of Vitamins in Intestinal Inflammation and Cancer

In intestinal tumors, numerous clinical studies had displayed a higher prevalence of CRC in the people with B vitamins and vitamin D deficiency [37,38]. Simultaneously, the prolonged unhealing of IBD puts patients at a higher risk of CRC. IBD patients with low vitamin D levels have worse disease severity and prognosis [39]. This shows that the role of vitamins in IBD and CRC cannot be ignored. In an inflammatory context, knockout of IKKbetaβ, a link between inflammation and cancer, reduces the occurrence of cancer due to an increased apoptosis of epithelial cells [40]. The presence of IL-6 trans-signaling increases the risk of inflammatory carcinogenesis in a study on colitis-associated premalignant cancer (CApC) [41]. As discussed above, intestinal inflammation and cancer are inextricably linked; moreover, intestinal inflammation caused by intestinal flora disturbance and vitamin deficiency may eventually develop into cancer if left untreated. In the following, we mainly discuss the role of vitamin deficiency in intestinal inflammation and carcinogenesis, and the possible mechanisms involved in producing these effects.

### 3.1. The Role of Vitamin A in Intestinal Inflammation and Cancer

There has been a surge in attention to the role of vitamin A in IBD. Vitamin A and its active metabolite retinoic acid (RA) play a pivotal role in the human immune system and may have an effect in the differentiation of helper T cells [42]. Under non-inflammatory conditions, RA is able to inhibit IL-6 receptor expression and Th1/Th17 production. Under inflammatory conditions, RA changes from a protective effect on the mucosa to a destructive effect; this is reflected in the significantly increased RA level in the mucosa of IBD patients at the active stage, accompanied by the up-regulation of IL-17 and IFN-γ secreted by CD4 and CD8 [43,44]. Vitamin A and its metabolites display an anti-inflammatory role by blocking the activation of Th1 and Th17, inhibiting the production of IL-17, INF-γ and TNF-α. Meanwhile, they can promote anti-inflammatory factors by cooperating with TGF-β to increase the level of Foxp3 to exert immune function [45]. One piece of data showed that low levels of vitamin A activate nuclear NF-kB and promote collagen formation; this exacerbates the inflammation of colitis. After vitamin supplementation, intestinal inflammation was obviously relieved [46]. All-trans retinoic acid (AtRA) reduces the expression of TNF-α and nitric oxide synthase 2 (NOS2) proteins secreted by the colonic mucosa of UC and CAC patients [47]. Vitamin A also has the effect of protecting the intestinal mucosal barrier, and the underlying mechanism is to antagonize the intestinal destruction effect of LPS [48,49].

In an examination on the effect of vitamin A deficiency on the development of colitis and CRC, researchers used dextran sodium sulfate (DSS) to induce colitis in mice; in addition, a combination of azoxymethane (AOM) pre-injection and DSS colitis induced CAC. Vitamin-deficient mice have higher levels of intestinal inflammation, slower mucosal healing, and enhanced immune responses more prone to CAC [50]. In the CAC mouse model, inflammation induced by gut bacteria affects AtRA metabolism; this leads to a decrease in its levels. The decreased activity of AtRA metabolic enzymes and decreased AtRA levels were found in clinical samples of UC and its associated CAC. Meanwhile, AtRA exerts anticancer effects by activating CD8^+^ T cells; this provides new insights for the treatment of CAC [51]. The combination of retinol and a retinol-binding protein (RBP) activates the oncogene STRA6 to promote the occurrence of CRC; the holo-RBP/STRA6 pathway can further play a carcinogenic role by promoting the carcinogenesis of fibroblasts [52]. In an animal experiment on the effect of vitamin A deficiency on the development of colitis and CRC, when vitamin A is at a low level, the vitamin A lipid droplets in mice will be degraded, the immune response will be enhanced, the colonic inflammation will be aggravated, and the progression of carcinogenesis will be accelerated [50].

### 3.2. The Role of Vitamin B12 and Folic Acid in Intestinal Inflammation and Cancer

The intestinal flora is involved in the human body’s metabolism of vitamin B12, which is also an important nutrient necessary for bacteria and enzymes in the human body [53]. There are many causes of vitamin B12 and folic acid deficiencies in IBD patients. These include microbial overgrowth in the ileum and jejunum, insufficient intake of vitamin B12 or increased body needs, increased intestinal destruction of vitamins or a decreased ability to absorb them, adverse effects of certain medications such as methotrexate or sulfasalazine, and some pathological causes, including protein-losing enteropathy, abnormal liver function, ileum-related lesions or surgical resection, intestinal fistula, etc. [54,55,56,57]. Unlike the well-established relationship between folate deficiency and IBD, many related studies on vitamin B12 and IBD have shown mixed results [54,58,59]. Vitamin B12 deficiency does not affect the healthy gut microbiota composition; however, it leads to a larger dysbiosis of the gut microbiota in experimental colitis and promotes the growth of opportunistic pathogens. Unexpectedly, vitamin B12 deficiency reduced colonic tissue damage; this is possibly related to an increase in the anti-inflammatory cytokine IL-10 [60].

A study was conducted on the potential role of a methyl-deficient diet (MDD), which reduces the plasma concentration of vitamin B12 and folic acid, and raises homocysteine levels, on the DSS-induced colitis in mice. DSS-treated mice fed with MDD had more serious colitis than the other treatment groups. Although superoxide dismutase and glutathione peroxidase activity remained stable, the levels of caspase-3 and Bax were affected. In addition to the increased expression of Bcl-2, the expression of inflammation-related markers, such as cytosolic phospholipase A2 and cyclooxygenase 2, had a significant trend of increasing; this was accompanied by the decreased expression of the tissue inhibitor of the metalloproteinase (TIMP) 3 protein. Therefore, vitamin B12 deficiency may aggravate the degree of inflammation in experimental IBD [61].

In CRC patients, long interspersed nuclear element-1 (LINE1) methylation in the tumor area and peripheral blood mononuclear cells (PBMCs) was demonstrated to be reduced in the high vitamin B12 group compared with the low serum vitamin B12 group; the tumor area is also lower in the LINE1 methylation level than the surrounding non-tumor area. Therefore, high vitamin B12 levels may reduce the risk of CRC by reducing DNA methylation [62]. Oxidative stress is one of the mechanisms of CRC pathogenesis; further, the levels of folate and vitamin B12 are positively correlated with the levels of the body’s antioxidant glutathione. Elevating folate and vitamin B12 levels in AOM-induced CRC showed significant anti-apoptotic, anti-oxidative stress, and resistance to AOM cytotoxicity [63]. In a systematic review and meta-analysis of 4517 IBD patients, folic acid supplementation was shown to reduce the CRC risk in IBD patients and to be protective against CRC development [64]. Interestingly, evidence shows that the lack of methyl donor nutrients folate, choline, methionine, and vitamin B12 inhibits tumor development in Apc mutant mice [65]. All in all, the role of vitamin B12 and folic acid in intestinal diseases requires more in-depth research.

### 3.3. The Role of Vitamin D in Intestinal Inflammation and Cancer

Epidemiological and animal experimentation have shown that vitamin D deficiency is a high-risk factor for IBD and CRC. Vitamin D supplementation contributes to reducing disease severity, possibly through a variety of mechanisms, including the regulation of immune cell trafficking and differentiation, and antimicrobial peptide synthesis [66]. Vitamin D can maintain the normal function of the intestinal mucosal barrier, and improve the body’s innate and adaptive immunity [67]. 1α, 25-dihydroxyvitamin D3 (calcitriol), the active form of Vitamin D, can combine with TGF-β, increasing IL2 levels, to modulate T cells to suppress inflammatory cytokine production and enhance the survival and function of Foxp3^+^ Treg cells [68]. The vitamin D receptor (VDR) is an important pathway for vitamin D to regulate immunity and play an anti-inflammatory role. Relevant data show that VDR has a protective effect on the gut; it may maintain gut homeostasis and prevent cancer by regulating the JAK/STAT pathway [69]. In patients with IBD, the content of VDR in the colonic epithelium was significantly lower than that of normal people. Among the experimental colitis models, transgenic mice expressing hVDR have less colonic inflammation than mice lacking VDR. Restoring epithelial VDR expression with hVDR transgene alleviates severe colitis and reduces mortality. The intrinsic mechanism is that the VDR exerts an anti-apoptotic effect by inhibiting NF-κB activation in order to protect the intestinal barrier to relieve colitis [70].

The expression level of the IBD risk gene ATG16L1 and the normal function of Paneth cells are closely related to VDR [71]. Paneth cell-specific VDR knockout (VDR^ΔPC^) mice exhibited a decrease in antibacterial activity, and an increased susceptibility to intestinal damage caused by *Salmonella* and the non-steroidal anti-inflammatory drug indomethacin. When VDR^ΔPC^ mice were co-housed with non-VDR-knockout mice, VDR^ΔPC^ mice exhibited resistance to DSS-induced colitis. Therefore, the VDR in Paneth cells has the ability to maintain antibacterial activity, thereby avoiding inflammation [72]. Paneth cell abnormalities and impaired autophagy can also be caused by an intestinal VDR deficiency [73]. 1,25-dihydroxyvitamin D3 increased the transcriptional and translational levels of Atg16L1 via VDR; this resulted in increased autophagy-related proteins, but restricted IL-1β expression [74]. Active vitamin D exerts anti-infective effects by regulating IL-8 and hBD-2 levels in IECs through PI3K/Akt signaling, and NOD2 protein expression after *Salmonella* invasion of epithelial cells [74].

A human trial published in 2018 displayed that vitamin D supplementation could reduce intestinal inflammation markers in patients with active UC [75]. Another mechanism study showed that VDR activates Claudin-15 (VDR’s target gene) to prevent colitis; this confirms the alleviating effect of vitamin D on IBD [76]. Simultaneously, the combined use of cyclosporin A and 1,25-Dihydroxyvitamin D3, or its non-hypercalcemia analogues EB 1089 and KH 1060, inhibits the proliferation of T lymphocytes in patients with active UC [77]. Numerous studies have proven that vitamin D and its metabolites directly inhibit the development and progression of various cancers; moreover, they have potential chemo-preventive effects [78,79,80]. A Western diet (WD) refers to a diet high in fat and low in calcium, vitamin D, and fiber. This diet exacerbated vitamin D deficiency by increasing the levels of the vitamin D catabolic enzyme cyp24a1. If the calcium and vitamin D levels in WD are restored to standard levels, colon inflammation may be reduced and cancer risk will be prevented [81]. In the CAC mouse model induced by AOM and DSS, dietary vitamin D3 supplementation showed excellent anticancer properties and reduced CCL20 levels. CCL20 is a potential therapeutic target in carcinogenesis. Vitamin D3 inhibits the carcinogenesis of colitis by reducing the expression of p38MAPK/NF-κB signaling mediated by CCL20 [82].

Some prospective studies have verified that the higher the vitamin D intake and blood 25(OH)D level, the lower the risk of CRC [83,84,85]. In a cohort study of 2809 IBD patients, vitamin D levels were inversely associated with cancer risk; for every 1 ng/mL increase in plasma 25-hydroxy vitamin D, the risk of CRC is reduced by 8% [86]. Abnormal activation of Wnt/β-catenin signaling pathway promotes the development of precancerous lesions into tumors [87]. By inhibiting the Wnt/β-catenin signaling pathway, active vitamin D metabolites can limit cancer cell proliferation and promote the transformation of CRC cells expressing VDR into the epithelial cells. In addition, the level of VDR in tumor stromal fibroblasts is directly proportional to the survival time of CRC patients [88]. Cyclin-dependent kinase inhibitors such as p21 and p27 can inhibit the cycle of cancer cells and induce apoptosis; they can also mediate the inhibitory effect of vitamin D on the growth of cancer cells [89]. RAS-activating mutations are one of the mechanisms of CRC pathogenesis, and low levels of vitamin D may be linked to mutations [90,91]. Urszula Dougherty et al. revealed that VDR inhibits inflammation-to-cancer transition by negatively regulating the RAS and EGFR pathways, which increases Snail1 expression and reduces VDR expression in CRC cells [92]. The function that vitamin D modulates gut microbiota plays a pivotal role in the treatment of CRC, mainly by regulating intestinal probiotics such as *Akkermansia muciniphila*; this protects the intestinal mucosal barrier [93].

Vitamin A and D are absorbed in the proximal and distal jejunum, respectively. Vitamin E and K are mainly absorbed in the ileum; vitamins produced by microorganisms are mainly absorbed in the colon. Vitamin deficiency aggravates intestinal inflammation and even promotes cancer through multiple mechanisms. In IBD and CAC patients, the activities of all-trans retinoic acid (AtRA) metabolizing enzymes and AtRA levels are reduced; this regulates the TLR4/NF-κB signaling pathway to reduce the expression of TNF-α and NOS2. Vitamin A and AtRA exert anti-inflammatory effects by blocking the activation of Th1 and Th17, inhibiting the production of IL-17, INF-γ, and TNF-α, and stimulating Foxp3 expression; this results in the differentiation of naive T cells into anti-inflammatory TREG cells. The lack of vitamin A and AtRA in the state of IBD will affect the occurrence of the above mechanisms and lead to the aggravation of inflammation. Vitamin D exerts anti-apoptotic effects by inhibiting NF-kB activation through its receptors. 1,25(OH)2D increases Atg16L levels via the vitamin D receptor (VDR), enhances autophagic capacity, and enhances innate immunity by modulating IL-8 and hBD-2 levels in IEC through PI3K/Akt signaling and NOD2 protein expression. The binding of vitamin D to TGF-β is directly involved in the regulation of regulatory T cells. Vitamin D exerts anticancer effects by inhibiting RAS, EGFR, Wnt/β-catenin, and p38MAPK/NF-κB pathways. P21/P27 mediates the inhibitory effect of vitamin D on cancer cells. High vitamin B12 levels may decrease IBD and CAC risk by reducing DNA methylation, oxidative stress, and inflammatory factor levels.

The intestinal absorption of vitamins and the mechanism of action of vitamins in inflammatory bowel disease (IBD) and colitis-associated colorectal cancer (CAC) are schematically shown in Figure 1.

## 4. Therapeutic Role of Vitamins and Gut Microbiota in IBD and CAC

### 4.1. The Role of Vitamin–Microbiota Interaction in IBD and CAC

The key role in the occurrence and pathogenesis of chronic IBD is the influence of microorganisms (especially symbiotic microbiota) on the host mucosal immune function [94]. Meanwhile, intestinal microbiota and chronic inflammation have been proved to be closely related to tumorigenesis [95]. Vitamins have the function of regulating gut microbiota and protecting the gut. Therefore, the interaction of vitamins and microbiota may have great potential in the treatment of IBD and CAC.

Vitamin A achieves the effect of treating UC by promoting mucosal healing, promoting the increase of ASCFA-producing related bacteria, and reducing the level of UC-related bacteria [36]. *Propionibacterium freudenreichii* ET-3 produces the precursor of vitamin K2, 1,4-dihydroxy-2-naphthoic acid (DHNA), which activates the aryl hydrocarbon receptor (AhR) to ameliorate colitis and modulate gut microbiota [96]. The deficiency of vitamin D increases the abundance of *Bacteroidetes*, *Proteobacteria* phyla, and *Helicobacteraceae* families, decreases the abundance of *Firmicutes* and *Deferribacteres* phyla, and also affects E-cadherin expression and reduces the number of tolerating dendritic cells [97]. However, the co-administration of vitamin D in the treatment of IBD with rifaximin can affect the gut microbiota and the efficacy of rifaximin [98]. Vitamin D promotes the growth of *A. muciniphila* to protect the intestinal mucosal barrier, and these effects are particularly important in combating the development of CRC [93]. Research shows that vitamin E and its metabolites have great potential in regulating gut microbiota, reducing inflammation, and inhibiting carcinogenesis [99]. Furthermore, vitamin Eδ-tocotrienol (δTE) and its metaboliteδTE-13′-carboxychromanol (δTE-13′) increased the abundance of *Lactococcus* and *Bacteroides* in the gut and inhibited the production of inflammatory factors [100].

### 4.2. The Role of Diet in IBD and CAC

Unreasonable dietary structure may be involved in the occurrence and development of IBD and CAC. A high-fat diet (HFD), low-fiber foods, and low-vitamin foods increase the risk of IBD and CAC [101,102]. Vitamin-rich fruits and vegetables, and a Mediterranean diet (MD) reduce inflammation levels, disease activity, improve quality of life in IBD patients, and reduce IBD incidence [103,104,105]. A vitamin-deficient Western diet (WD) aggravates CAC symptoms and promotes tumor development; this is possibly mediated by STAT3 and NF-kB [106]. HFD can also promote the development of CAC by evading ferroptosis [107]. However, eating a MD rich in vitamins can counteract the gut damage caused by the HFD and inhibit tumor development [108]. The incidence of CRC decreases with increased MD adherence, and high compliance improves the body’s antioxidant capacity [109,110].

### 4.3. The Role of Probiotics That Produce Vitamins in IBD and CAC

Probiotics that produce vitamins have received increasing attention in recent years. The reason is that probiotics are effective in reducing the side effects caused by drugs. Among them, *lactic acid bacteria* (LAB) are the most concerned. LAB inhibit the inflammatory process through different mechanisms, including regulating the intestinal flora disorder of patients with IBD, protecting the normal function of the intestinal barrier and mucosa, and regulating the immune response of human. LAB exert anti-inflammatory and antioxidant effects by producing riboflavin (vitamin B2) and folic acid [111,112]. These results have also been verified in animal experiments. Vitamin-producing LAB not only play an anti-inflammatory role in acute enteritis, but can also effectively relieve recurrent colitis. In addition, in the process of combined use with mesalazine, they can effectively reduce the adverse effects and improve the curative effects [113]. Mayur Garg et al. found that the injection of LAB that produce folic acid will reduce diarrhea in mice with 5-FU-induced enteritis, and improve the structure and function of colon tissue. This discovery reduces the severity of intestinal mucosal inflammation that occurs during cancer chemotherapy and improves drug effectiveness; thereby, this improves the quality of life of patients [114]. In addition, the combined use of LAB and 5-FU reduces the decrease in blood cell count caused by 5-FU and allows patients to obtain a complete treatment cycle [115]. Nahla M. Mansour et al. isolated three probiotics that produce riboflavin and folic acid from 150 collected human fecal samples; they used them to treat acetic acid-induced colitis in rats. They found that these probiotics protect the colonic mucosa and promote the healing of ulcerative lesions; moreover, they have anti-inflammatory and anti-oxidative stress effects [116]. A newly isolated bacterium with the ability to produce folic acid, *Latilactobacillus sakei* LZ217, has the function of promoting the production of butyric acid and improving the composition of intestinal flora [117]. *Akkermansia muciniphila*, a common bacterium in the gut, regulates CLTs to protect the gut against inflammation and tumor invasion; it also produces vitamin B12 to alleviate vitamin deficiency in IBD patients [118,119]. Studies have found that the *Propionibacterium* strain, P. UF1, synthesizes vitamin B12; this has a positive regulatory effect on intestinal immunity and intestinal health [120].

Similarly, *E. coli* alleviates IBD by producing vitamins. Lesley Wassef et al. and Jennifer K. Miller et al. used *E. coli* to produce two β-carotene-producing strains to treat vitamin A deficiency. These results show great clinical potential [121,122]. The combination of Vitamin A and its metabolites with *Lactobacillus brevis* KB290 raises the CD11c^+^ MP/CD103-DC ratio; thereby, this plays a positive role in the treatment of colitis [123]. Furthermore, segmented filamentous bacteria (SFB) in the gut can produce AtRA to counteract damage to the gut from infection [124]. Probiotics have a positive effect on vitamin D and its receptor activity, such as *Lactobacillus rhamnosus* GG (LGG) and *Lactobacillus plantarum* (LP); also in the *Salmonella* colitis model, using VDR (-/-) mice to verify that the alleviating effect of LGG on IBD is through the VDR signaling pathway [125]. In addition, bile salt hydrolase (BSH)-active *Lactobacillus reuteri* NCIMB 30,242 modulates active vitamin D levels in plasma. A mixture of krill oil (KO), the probiotic *Lactobacillus reuteri*, and vitamin D significantly reduced pathological scores and the release of inflammatory factors, and promoted mucosal healing and reduced the occurrence of opportunistic infections [126]. The level of VDR is significantly increased after pretreatment with the probiotic VSL#3, which together protect intestinal mucosa and prevent damage; this plays a certain role in preventing the development of CRC [127]. Lu, R. et al. discovered that after treating HCT116 cells or intestinal organoids with a conditioned medium of LAB isolated from Korean kimchi, its secreted proteins P40 and P75 are related to the increased expression of VDR; they also enhance the autophagy response, which together has an anti-inflammatory effect [128]. Lithocholic acid (LCA) synthesized by gut microbes acts as a bridge to link VDR to microbes, thereby increasing vitamin D levels [129].

The role of vitamin-producing probiotics in intestinal inflammation and carcinogenesis is shown in Table 2.

### 4.4. The Role of Fecal Microbiota Transplantation (FMT) in IBD and CAC

FMT has great potential as a strategy for the treatment of IBD. FMT alleviated the symptoms of IBD, alleviated intestinal damage, and reduced disease activity by modulating immunity, inhibiting the production of inflammatory factors, and reducing the level of oxidative stress. FMT can also increase bacterial diversity and improve the dysbiosis of the IBD patient’s gut microbiota. At the genus level, the abundances of *Lactobacillus*, *Faecalibacterium*, *Butyricicoccus*, *Blautia*, *Lachnoclostridium*, *Coriobacteria*, *Olsenella*, *Bifidobacterium*, and butyric acid-producing bacteria were increased after FMT; in addition, the numbers of *Clostridium*, *Bacteroides*, and *Helicobacter* were decreased [130,131,132,133,134]. FMT has a therapeutic effect on CAC by regulating Treg cells, reducing the level of inflammatory factors, and regulating the changes of flora. FMT increased *Firmicutes* abundance and decreased *Bacteroidetes* abundance, restoring gut microbiota to normal levels [135]. Interestingly, some gut microbes that are altered after FMT treatment had the ability to synthesize vitamins [28,29]. Moreover, the role of vitamin A in protecting the gut by regulating gut flora has also been verified by FMT [36,136]. Therefore, FMT not only directly inhibits intestinal inflammation and modulates the microbiota and immune system, in order to treat IBD and CAC, but may also assist in the treatment of IBD and CAC by transplanting bacteria with the ability to synthesize vitamins.

The interaction mechanism of vitamins and gut microbiota in inflammatory bowel disease (IBD) and colitis-associated colorectal cancer (CAC) is schematically shown in Figure 2.

Microbiota dysbiosis and vitamin deficiencies exacerbate the occurrence and development of IBD and CAC. Gut microbiota can influence vitamins synthesis, metabolism, and transport. Vitamin supplementation can affect the metabolism and composition of intestinal flora, and increase alpha diversity, increase beneficial bacteria, and reduce pathogenic bacteria. Diet, probiotics/prebiotics, and fecal microbial transplantation (FMT) can not only regulate the gut microbiota, but also participate in the absorption and metabolism of vitamins, and improve vitamin deficiencies.

## 5. Conclusions

Vitamins act as powerful signaling molecules through different nuclear receptors and cell signaling pathways to play an anti-inflammatory role and inhibit tumor development. Disorders of the intestinal microbiota can not only induce IBD by promoting inflammation and affecting the production of metabolites, but also by mediating DNA damage, inducing specific signal pathways; in turn, this promotes immune cell infiltration and blocks anti-tumor immunity to affect the intestinal inflammation state and pretumor environment to exert a carcinogenesis effect. Furthermore, the destruction of the intestinal microbiota influences the absorption and production of vitamins, and aggravates intestinal diseases. This review discusses the metabolism of vitamins and their interaction with the flora; it focuses on the effects of various vitamins on intestinal inflammation and tumorigenesis, and the possible mechanisms involved.

Our knowledge of the immune regulation and anti-inflammatory mechanism exerted by vitamins is still superficial. Given that IBD and CAC patients generally suffer from vitamin deficiency and intestinal flora disorders, targeting these phenomena to improve intestinal inflammation and inhibit tumorigenesis may provide possibilities for the prevention and treatment strategies of IBD and CAC. The high cost, high toxicity, and undesirable adverse side effects faced by current IBD therapies are the direct reasons for the continuous development of new therapies. In this regard, newly developed fecal transplants and probiotic supplements show some value in their use as adjunctive treatments for IBD, and in reducing harmful side effects from primary treatments [137]. There is currently great interest in the use of probiotics for the treatment of gastrointestinal diseases. Some of these beneficial bacteria not only have innate immune regulation properties, but also produce large amounts of vitamins; this helps prevent malnutrition and provides antioxidant and anti-inflammatory effects. However, the safety and effectiveness of its application to the human body still need to be explored; further, many innovative and in-depth tests need to be carried out on human patients to overcome the limitations of probiotic therapy. In general, vitamin-producing probiotics have broad prospects in the treatment of IBD and prevention of cancer; thus, they own great clinical value.

## Figures and Tables

**Figure 1 nutrients-14-03383-f001:**
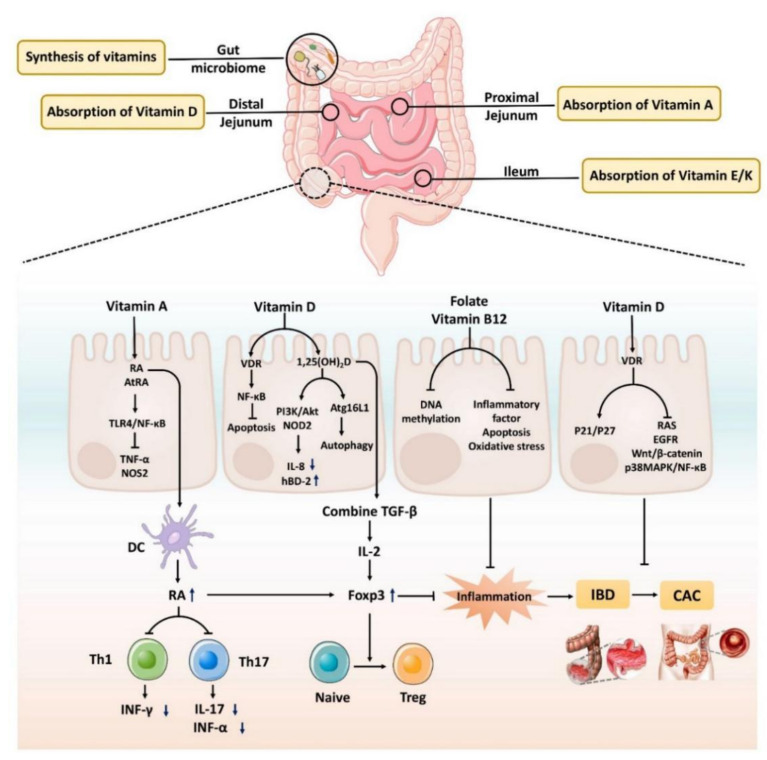
Intestinal absorption of vitamins and the mechanism of action of vitamins in inflammatory bowel disease (IBD), and colitis-associated colorectal cancer (CAC). The up and down arrows in Figure 1 represent increase and decrease, respectively. DC represent dendritic cell, which is the most functional professional antigen-presenting cells in the body. RA stands for retinoic acid.

**Figure 2 nutrients-14-03383-f002:**
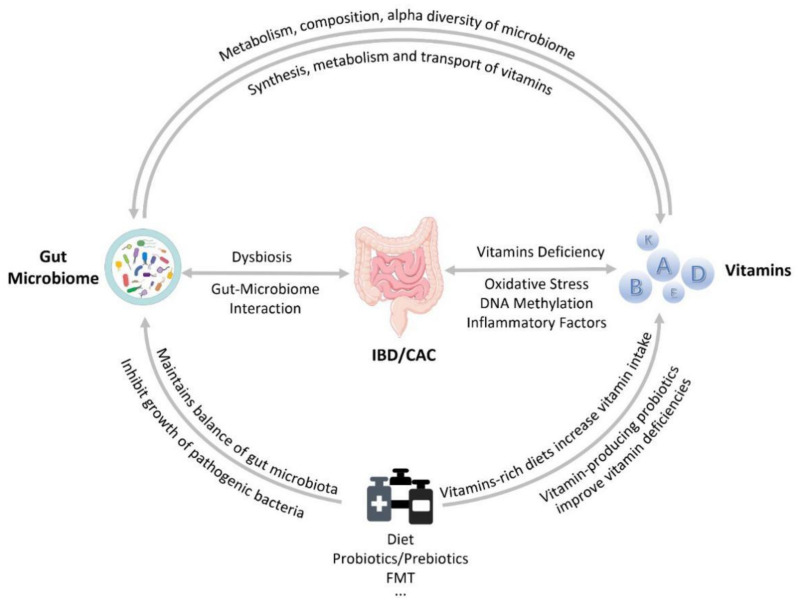
The interaction mechanism of vitamins and gut microbiota in inflammatory bowel disease (IBD) and colitis-associated colorectal cancer (CAC). Fecal microbiota transplantation, FMT, is defined as the transplantation of functional flora in the feces of healthy people into the gastrointestinal tract of patients to rebuild new intestinal flora and achieve the treatment of intestinal and extra-intestinal diseases.

**Table 1 nutrients-14-03383-t001:** Effects of vitamin supplementation on the human gut microbiota. The table is modified according to: [15,30,32,33,34,36].

Vitamin Supplement	Increased Bacteria	Reduced Bacteria
Vitamin A	*Akkermansia, Lactobacillus, Prevotella, Aerococcus*	*Bacteroides, Parabacteroides, Escherichia/Shigella, Klebsiella, Oscillibacter,* *Pseudolavonifractor, Clostridium sensu stricto, Butyrimimonas, Mucispirllum, Clostridium XIVb* [36]
Vitamin B	*Actinobacteria, Odoribacteraceae*	*Campylobacteraceae,* *Fusobacteriaceae,* *Prevotellaceae* [15]
Vitamin C	*Lactobacillus sp.*	*Enterobacteriaceae* [32]
Vitamin D	*Actinobacteria, Prevotella*	*Bacteroidetes,* *Veillonella,* *Haemophilus, Coprococcus, Bifdobacterium* [30,33]
Vitamin E	*Bacteroides,* *Proteobacteria*	*Ruminococcus, Lachnospiraceae, Muribaculaceae* *(In the case of vitamin E deficiency)* [34]

**Table 2 nutrients-14-03383-t002:** The role of vitamin-producing probiotics in intestinal inflammation and carcinogenesis. The table is modified according to: [111,112,116,117,118,119,120,121,122,124,125,126,127,128].

Probiotics	Product	Effect
*Lactic acid bacteria*	Vitamin B2 Folic acid	Anti-inflammatory and antioxidant [111,112]
*Pediococcus acidilactici*	Vitamin B2 Vitamin B9	Colonic mucosal protection and promotion of healing of ulcerative lesions [116]
*Latilactobacillus sakei LZ217*	Folic acid	The role of butyric acid production and improvement of intestinal flora composition [117]
*Akkermansia muciniphila*	Vitamin B12	Regulation of CLT to protect the gut from inflammation and tumor invasion [118,119]
*Propionibacterium strain, P. UF1*	Vitamin B12	Intestinal immune regulation [120]
*Escherichia coli MG1655*, Nissle 1917 (EcN-BETA)*	Vitamin A	Treating vitamin A deficiency [121,122]
*Segmented filamentous bacteria*	All-trans retinoic acid	Counteracts damage to the gut from infection [124]
*Lactobacillus rhamnosus GG and Lactobacillus plantarum*		Alleviation of IBD through the VDR signaling pathway [125]
*Lactobacillus reuteri NCIMB 30242*		Regulation of active vitamin D levels in plasma [126]
*Probiotic VSL#3*		Increase VDR levels and prevent CRC [127]
*Lactic acid bacteria (Isolated from Korean kimchi)*	P40 P75	Increase VDR expression and enhance autophagic response [128]

## Data Availability

Not applicable.

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
