# Peer review of "Vitamin–Microbiota Crosstalk in Intestinal Inflammation and Carcinogenesis"

_nutrients, 2022, doi:10.3390/nu14163383_

Round 1
Reviewer 1 Report
Despite the idea of the paper is interesting, I have major concerns regarding this manuscript.
First of all, the Authors should perform a systematic review to show to the readers the current knowledge of this very complex issue. The methods of search of the literaturÄ™ databases should be shown.
It is very difficult to follow the paper. It consists of many parts, that do not have really a good connection with each other. Section „Conclusions” is to long.
I can not absolutelu agree with the Authors about this part of the manuscript :
According to this aspect, newly developed fecal transplants and probiotic supplements have shown the desired efficacy and can be used as an adjuvant treatment of IBD to reduce harmful side effects caused by the main treatment. Among them, LAB are currently the most commonly used probiotics, which are used to treat various inflammatory diseases, especially gastrointestinal diseases, and show excellent therapeutic effects.
Probiotics are not the standard therapy for IBD! The evidence of the effectiveness of probiotics in GI diseases is limited. The paper CANNOT conclude, that they have an excellent therapeutic effect! It is misleading.
There are also many minor issues:
1. The inflammation is not limited to the mucosa in case of CD. (line 28)
2. Gut microbiota disturbance is one of the common pathogenesis of IBD and CRC. (line 70) What does this sentence means?
3. Lines 123-128 – very general information situated in the wrong section
4. Lines 422-423 – very general information, basic knowledge and not the conclusion of the paper
Reviewer 2 Report
The current review attempts to summarize the literature on vitamins and intestinal inflammation. The manuscripts presented however, do not appear to me collected systematically. The topic is of interest, however, is written, perhaps more as a perspectives, opinion, or state-of-the-art review. Some of the text is very lengthy, and could benefit from further summarization, as tables perhaps.
It would be helpful if author could:
1-define the review
2-explain how the articles were amassed
It is challenging not to infer potential author bias.
There are some issues with the English:
Abstract:
Line 12-13: should read, “Besides, intricate relationships exist between vitamins and gut microbiota.” This sentence is a bit awkward, consider revising.
Main text:
Line 58: should read, “More and more evidence shows that vitamin deficiency…” some of the subject-verb agreement is not.
Line 61: vitamins should be singular “vitamin”.
Section 2. could some of this be summarized as a table, as there is a lot of text.
Line 124: should read, “Humans obtain vitamins…”
Figure 1: is this an original image for this review, if not it should be cited.
Check lines 233-235 and 297, is there some repetition here, is this intentional? Similarly, lines 206-207 and 303-305.
Generally, speaking there are some organizational issues in the manuscript that could be improved. I.e. section 3.3 is for Vitamin D, and yet on Line 286, there is mention of Vitamin A & D. This being said there is a fair amount of “back-n’-forth” between epidemiology, expression, and mechanisms of action, which is a bit confusing for the reader.
Section 4.3 could be summarized as a table.
Line 438: the word “shallow” could be substituted with “superficial” or “limited”.
There are sufficient abbreviations to warrant an abbreviations section.
Round 2
Reviewer 1 Report
Dear Authors, thank you for following the remarks and improvements to the paper. Yet, I would like to ask for the providing more clear subsections as the paper is large and difficult to follow.
Reviewer 2 Report
The authors need to define the review, this was not addressed, ie. narrative, state-of-the-art, vs. systematic.
This is where the protocol is drafted, registered, and a PRISMA chart is generated. Often involving the librarian support, who is not directly involved with the project furthering the chance of author bias. If authors could provide a more detailed methodology of search strategy.
